# Sex differences in body composition but not neuromuscular function following long-term, doxycycline-induced reduction in circulating levels of myostatin in mice

**Dallin Tavoian[1], W. David Arnold[2], Sophia C. Mort[1], Sonsoles de Lacalle[3]¤ \***

**1** Program in Translational Biomedical Sciences, 1 Ohio University, Athens, OH, United States of America,
**2** Departments of Neurology, PM&R, and Neuroscience, and Physiology and Cell Biology, The Ohio State University, Columbus, OH, United States of America, **3** Sonsoles de Lacalle, Department of Biomedical Sciences,1 Ohio University, Athens, OH, United States of America

¤ Current address: Department of Health Science, California State University Channel Islands, One University Drive, Camarillo, CA.
\* sdelacalle@mac.com

**Data Availability Statement:** All relevant data are within the article and its Supporting Information

## Abstract

Age-related declines in muscle function result from changes in muscle structure and contractile properties, as well as from neural adaptations. Blocking myostatin to drive muscle growth is one potential therapeutic approach. While the effects of myostatin depletion on muscle characteristics are well established, we have very little understanding of its effects on the neural system. Here we assess the effects of long-term, post-developmental myostatin reduction on electrophysiological motor unit characteristics and body composition in aging mice. We used male (N = 21) and female (N = 26) mice containing a tetracycline-inducible system to delete the myostatin gene in skeletal muscle. Starting at 12 months of age, half of the mice were administered doxycycline (tetracycline) through their chow for one year. During that time we measured food intake, body composition, and hindlimb electromyographic responses. Doxycycline-induced myostatin reduction had no effect on motor unit properties for either sex, though significant age-dependent declines in motor unit number occurred in all mice. However, treatment with doxycycline induced different changes in body composition between sexes. All female mice increased in total, lean and fat mass, but doxycycline-treated female mice experienced a significantly larger increase in lean mass than controls. All male mice also increased total and lean mass, but administration of doxycycline had no effect. Additionally, doxycycline-treated male mice maintained their fat mass at baseline levels, while the control group experienced a significant increase from baseline and compared to the doxycycline treated group. Our results show that long-term administration of doxycycline results in body composition adaptations that are distinctive between male and female mice, and that the effects of myostatin reduction are most pronounced during the first three months of treatment. We also report that age-related changes in motor unit number are not offset by reduced myostatin levels, despite increased lean mass exhibited by female mice.

files. The table is included at the end of the article, and a separate file has been added too.

**Funding:** This work was partially funded through the National Institutes of Health Grant number R03AG050877 to WDA, and a Vision 2020 Award from the Osteopathic Heritage Foundations to SL, SCM and DT. The funders had no role in study design, data collection and analysis, decision to publish, or preparation of the manuscript.

**Competing interests:** The authors have declared that no competing interests exist.

## Introduction

Functional limitations that accompany age-related loss of muscle mass and strength are a major public health concern, affecting 10% of the world population over the age of 60, and as much as 50% over the age of 80 [1]. As human longevity increases, the number of years that individuals live with sarcopenia and physical disability also increase, affecting females disproportionally [2–4]. Sarcopenia is characterized by reduced muscle quality, mass, strength, and power, and is associated with increased risk of falls and mortality [5]. In the past, declines in strength and power were believed to be directly dependent on changes in muscle mass [6]. However, recent reports indicate that the association between loss of strength and mass is low [7–10], suggesting non-mass dependent contributors. One such contributor is impairment of the motor unit (defined as a motor neuron and the muscle fibers it innervates [11]). Humans begin to lose motor units after the age of 60, and those that remain show changes in morphology, behavior, and electrophysiology, accompanied by impaired muscular performance and decreased muscle mass [12]. Sarcopenia research is mostly focused on muscle mass and strength, but the motor unit does not receive adequate consideration.

Myostatin (MSTN), a negative regulator of muscle mass, is a protein synthesized by the Growth Differentiation Factor-8 gene, or *Mstn* gene, and is a member of the Transforming Growth Factor-β superfamily [13]. Constitutive (i.e., permanently inactivated) deletion of the *Mstn* gene in mice ($Mstn^{-/-}$) results in up to a 160% increase in skeletal muscle mass from hypertrophy and hyperplasia [13–16], suggesting that MSTN reduction has a strong therapeutic potential to treat age-related muscle wasting in humans. However, only a small proportion of studies on the physiological effects of MSTN reduction have included neurological measures [17,18]. These few studies have reported increased axonal number, motor unit number, and motor unit size [17,18], as well as resistance to age-related axonal loss in $Mstn^{-/-}$ mice [17]. Because these studies were performed in constitutive knock-out mice, it is not known whether post-developmental MSTN reduction would have the same neuroprotective effect. Furthermore, skeletal muscle adaptations are more extensive when MSTN is reduced during development than when it is reduced in adulthood [13–16,19–24].

Methods to manipulate MSTN levels in the fully-developed organism include the use of inhibitory MSTN antibodies [25] and propeptides [26] in wild type (WT) mice, as well as *Mstn* gene excision induced in transgenic mice with a tamoxifen- or doxycycline (DOX)- inducible Cre recombinase mutation [20,22]. These approaches, which more closely mimic the treatments to be used in the target population (i.e., MSTN reduction in older adults) [25,27–30], result in skeletal muscle hypertrophy ranging from 15–45% over controls, but without the hyperplasia or fiber-type shifts seen in $Mstn^{-/-}$ mice [23,26]. It is encouraging that hypertrophy can be induced in old mice (18–24 months of age) [25,27,30], but because these were short-term investigations (3 months or less), they do not provide information on the effective age to initiate treatment or the ideal duration for such treatment to elicit the maximal therapeutic effect. Furthermore, the use of these techniques is not exempt from pitfalls, including the potential confounding effects of the pharmacological agents used to drive gene expression [31,32].

We have a long-standing interest in understanding whether an extended reduction in MSTN, starting in adulthood, would counteract the effect of aging on neuromuscular characteristics and body composition in male and female mice. Our study was well underway when we became aware of the metabolic effects of DOX [32–34]. Although those reports are focused on cellular systems, we recognized that DOX itself could also have a measurable influence at the whole organism level. Thus, we decided to complete the project as designed, and analyze the results taking into consideration that DOX itself would have an impact beyond activating

the tetracycline-inducible (Tet-ON) system. Here we present the effects of administering DOX for 12 months to induce a reduction in MSTN levels starting at one year of age, in both male and female mice. During that time we collected weekly food intake and monthly body composition, as well as hindlimb electromyographic (EMG) recordings at 12, 15, 21, and 24 months of age. Our data suggests a sexual dimorphic response to long-term DOX administration. In addition, this time course also documents the progression of neuromuscular impairment due to aging, providing a possible window in the aging process in which an intervention would be most effective.

## Materials and methods

### Experimental subjects and design

Data was collected from 47 mice, 21 male and 26 female, the result of breeding the B6;129S7-$Mstn^{tm1Swel}$/J strain (Jackson Laboratories stock number 012685) with the B6;C3-Tg(ACTA1-rtTA,tetO-cre)102Monk/J mutant mouse strain (Jackson Laboratories stock number 012433). The strain number 012685 contains a targeted mutation with exon 3 of the *Mstn* gene flanked by loxP sites. The strain number 012433 carries a tetracycline (doxycycline) inducible Cre-mediated recombination system specific for skeletal muscle [35]. Two transgenic constructs were co-injected to generate this strain. The first transgene contains Cre recombinase under the control of the tetO, tetracycline-responsive regulatory element, and a second transgenic construct contains the reverse tetracycline-controlled transactivator, rtTA, under the control of the human ACTA1 (actin alpha1) skeletal muscle promoter. Treatment with DOX renders the *Mstn* gene non-functional by excising the DNA flanked by the loxP sequences. Further details on the development of these strains are provided by Jackson Laboratories. Tail snips collected at weaning were analyzed by Transnetyx Laboratories (Transnetyx Inc., Memphis, TN) to confirm genotype. Only mice with the homozygous floxed genotype, that is, testing positive for the *tTA*, *CRE*, and *MSTN-3 FL* probes, and negative for the *MSTN-3 WT* probe, were used in this study. For the sake of clarity, we refer to the experimental group of mice that received DOX as the "treated" group, and to the experimental group that did not receive DOX as the "control" group. An additional group of control mice (n = 7, all male) that tested negative for the Cre gene were kept in the cage with the DOX group are labeled "Cre⁻ Treated" and a second group of control mice (n = 2, all male) without the Cre transgene are labeled as "Cre⁻ Control".

All mice were housed in groups of 2–4, kept on a 12:12 light:dark cycle, with food and water *ad libitum*, using the standard RMH-3000 chow. Following baseline measurements performed between 50 and 52 weeks (12 months) of age, mice were randomly assigned to either the regular chow (n = 10 male, n = 12 female), or to a special diet (Harlan's Teklad 8640) containing 200 mg/kg of DOX (n = 11 male; n = 14 female), maintained until the end of the study. Teklad 8640 and RMH3000 have identical protein and fat levels (22% protein, ~5% fat) and contain similar ingredients (wheat, soybean meal, corn, wheat middlings, and fish meal). The following measures were collected throughout the study: amount of food consumed, body weight, body composition, and neuromuscular physiology. Mice were euthanized by $CO_2$ terminal anesthesia followed by cervical dislocation at age 24 months.

This study was carried out in accordance with the recommendations in the Guide for the Care and Use of Laboratory Animals of the National Institutes of Health, following a protocol approved by Ohio University's Institutional Animal Care and Use Committee. All efforts were made to minimize suffering.

## Serum myostatin measurements

At the time of euthanasia, blood was collected from 21 treated (9 male and 12 female) and 19 control (8 male and 11 female) mice, and total serum MSTN concentration was measured with the GDF-8/Myostatin Quantikine® ELISA kit (Catalog Number DGDF80, R&D Systems, Minneapolis, MN, USA) according to the manufacturer's instructions. Total MSTN includes both the mature protein and pro-myostatin. Pro-myostatin was converted to the immunoreactive form using a 10-minute acid activation with 1 mol HCl per liter of water. Thus, both active (mature) and latent (pro-peptide) MSTN concentrations were included in the total concentration determined by the assay [36,37]. Measurements were performed in triplicate for each sample with the exception of three female control samples, which were performed in duplicate.

## Body composition

Animal weights were recorded weekly. Food was weighed at initial provisioning and weekly thereafter before supplementing with a known amount. A weekly estimate per animal was calculated as the total amount of food consumed each week divided by the number of animals in the cage. *In vivo* measurements of body composition were taken monthly using a Bruker Minispec (The Woodlands, TX, USA). The Minispec uses nuclear magnetic resonance technology to provide estimates of fat mass, lean mass (muscle and bone), and free body fluid for each animal [38].

## Electromyography

Compound muscle action potential (CMAP) and average single motor unit potential (SMUP) amplitudes were recorded, as we have previously described, from the triceps surae muscle at 12 (baseline), 15, 21, and 24 months of age with a portable electrodiagnostic system (Synergy EMG) [39–41]. Briefly, mice were anesthetized with an isoflurane vaporizer (Vetamac Inc., Rossville IN) and placed on a thermostatic warming plate. Veterinarian petroleum-based jelly was applied to the eyes to prevent dryness. Hair was removed from the sacral region and the right hind limb using animal hair clippers to allow anatomical visualization, placement of stimulation and recording electrode, and to reduce signal interference. CMAP responses were recorded using a pair of surface recording ring electrodes (Alpine Biomed) placed over the triceps surae and over the metatarsal region of the foot following supramaximal stimulation of the sciatic nerve (<10mA) via a pair of monopolar needle electrodes (Teca). A ground electrode (Carefusion) was placed on the tail. Ten all-or-none incremental responses were obtained following submaximal stimulations of the sciatic nerve. These incremental responses were averaged to determine the average SMUP size (in amplitude). MUNE was calculated by dividing the peak-to-peak CMAP amplitude by the average SMUP amplitude (MUNE = CMAP/SMUP). An in-depth explanation of this protocol and list of equipment and supplies have been published previously [40]. Following data collection, the mouse was placed in an empty cage until fully recovered from anesthesia and then returned to its home cage.

## Data analysis

All data, expressed as percent change from baseline unless otherwise stated, were analyzed using SPSS version 25.0. A two-way ANOVA was run to examine the effect of sex and treatment on serum MSTN concentrations at 24 months of age. Two-way repeated measures ANOVAs were run to examine the effects of treatment and time on body composition. Separate analyses were run for males and females to identify sexual dimorphic effects of altered MSTN

levels on body composition [42,43]. Separate independent samples T-tests for males and females were run to compare average weekly food intake between 12 and 24 months of age for DOX-treated and controls groups. Because the effect of age, sex, and MSTN manipulation on motor unit characteristics has not been previously documented, we ran three-way repeated measures ANOVAs to dissect the contribution of each variable on motor unit characteristics. In those cases in which an interaction was identified, Bonferroni's *post-hoc* test for pairwise comparisons was applied. The Huynh-Feldt correction was used if the assumption of sphericity was violated. Statistical significance was set at 0.05. Raw data for all measurements (mean ± SEM) are presented in S1 Table.

## Results

### Serum myostatin concentration

Administration of DOX reduced MSTN levels in serum as expected, and we found a significant main effect of treatment, with lower amounts in DOX-treated (7.12 ± 0.8 ng/mL; mean ± SEM) than in controls (11.09 ± 1.1 ng/mL), a 37% reduction, $F(1,35) = 9.185$, $p = 0.005$. There were no differences between male and female mice ($p = 0.75$), and serum MSTN concentration was not significantly altered by the combination of sex and DOX treatment ($p = 0.61$).

### Time course of changes in body composition

**Female mice.** Normalized to body weight, DOX-treated females ate significantly less (75% of body weight) than their controls (84% of body weight) on average weekly, $t(26) = 2.21$, $p = 0.036$ (Fig 1B). Despite this difference in food consumption, both groups increased equally in total body mass along the 12 months of the study, remaining significantly higher than 12-month baseline at all time points (all $p$-values < 0.001), with no effect of DOX treatment $F(2.74, 65.86) = 0.451$, $p = 0.45$ (Fig 2A). Female lean mass increased only up to 18 months of age, at which point it steadily declined until 24 months of age, though it never dropped below the 12-month baseline values (Fig 2B). There was a statistically significant two-way interaction of time and treatment on lean mass, $F(2.86, 68.60) = 5.89$, $p = 0.001$. Follow-up comparisons applying Bonferroni correction showed that the increase in lean mass was significantly different from baseline at all time points (all $p$-values < 0.003). The increase in lean mass was more pronounced in DOX-treated mice than controls at 15 months of age (3 months of treatment) (mean difference 5.3%; 95% CI, 2.4% to 8.1%, $p = 0.001$), and the two groups continued to diverge, so that by 24 months of age DOX-treated mice had increased their lean mass 8.8% above the control group (95% CI, 3.7% to 13.8%, $p = 0.002$). Both treated and control groups of female mice also significantly increased their fat mass (Fig 2C) between the 12-month baseline and 15, 18, and 21 months but the difference disappeared by 24 months ($p = 0.024$) (all $p$-values < 0.005, Bonferroni corrected significance set at 0.0125). Fat mass in female mice was not modified by DOX treatment, $F(2.78, 66.76) = .151$, $p = 0.16$.

**Male mice.** DOX-treated male mice consumed slightly more food than controls, though it did not reach statistical significance, $t(12) = -1.928$, $p = 0.08$ (Fig 1A); when normalized to body weight, both groups ate similar amounts, 88% of body weight/week for treated and 78% of body weight/week for controls, $t(12) = -1.421$, $p = 0.18$ (Fig 1B). As with the females, there was no interaction effect of time and treatment on total body mass, $F(2.47, 46.93) = 0.175$, $p = 0.88$, and both treated and control male groups increased their total body mass along the 12 months of the experiment (Fig 3A), an effect that was significantly different from the 12-month baseline values at all time points measured (all $p$-values < 0.003). Lean mass (Fig 3B) was also significantly higher than the 12-month baseline at all time points (all $p$-

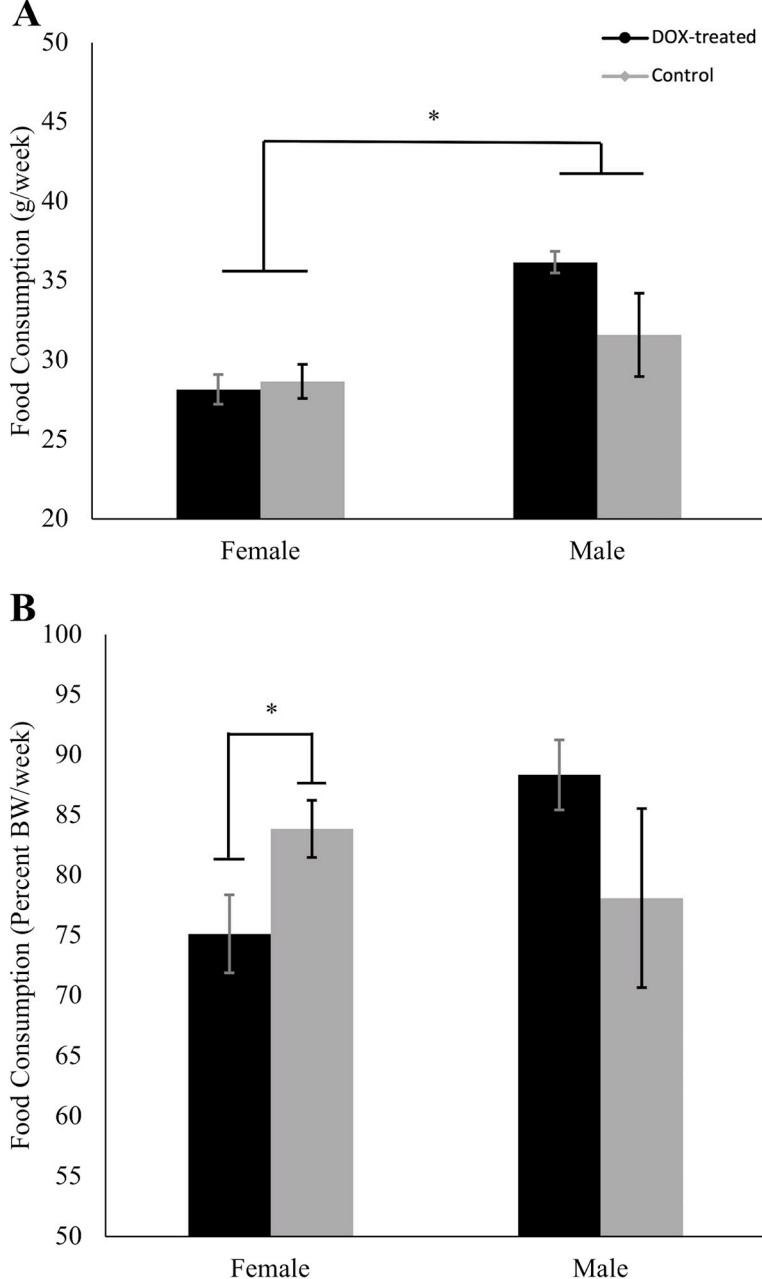

**Fig 1. Average weekly food consumption between 12 and 24 months.** (A) When absolute food consumption was compared between treated and control mice there was no effect of DOX treatment, but males ate significantly more than female mice. (B) When food consumption was normalized to body weight, DOX-treated females consumed less food than female controls, and there was no difference between DOX-treated males and controls. $^*p < 0.05$. Values are mean ± SEM. g, grams; BW, body weight.

values $< 0.001$). This increase was very pronounced at 15 and 18 months of age, remained elevated at 21 months, and declined at 24 months, remaining above baseline levels. In contrast to females, the difference in lean mass between DOX-treated and control mice shown in Fig 3B did not reach statistical significance, $F(3.54, 67.19) = 2.219$, $p = 0.08$.

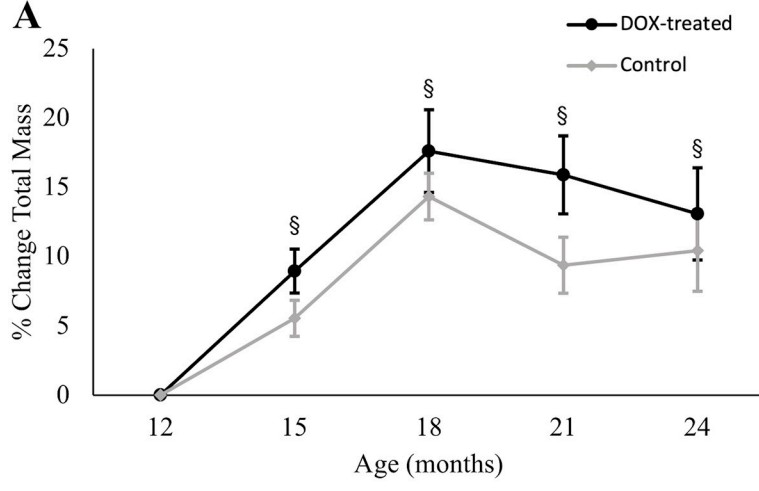

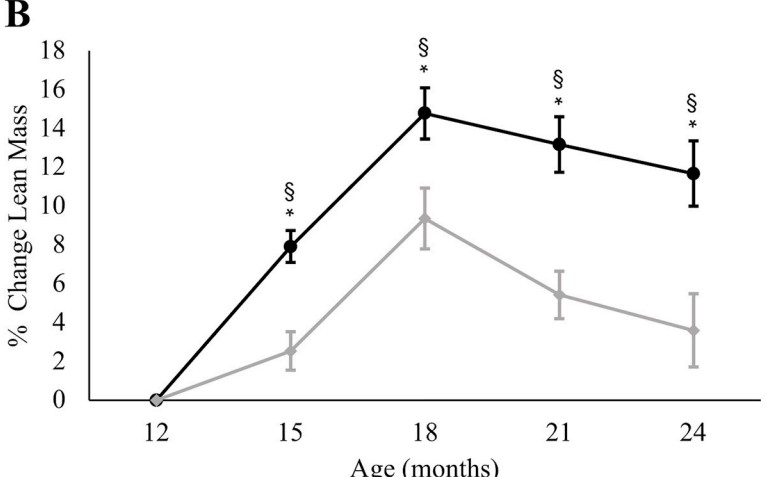

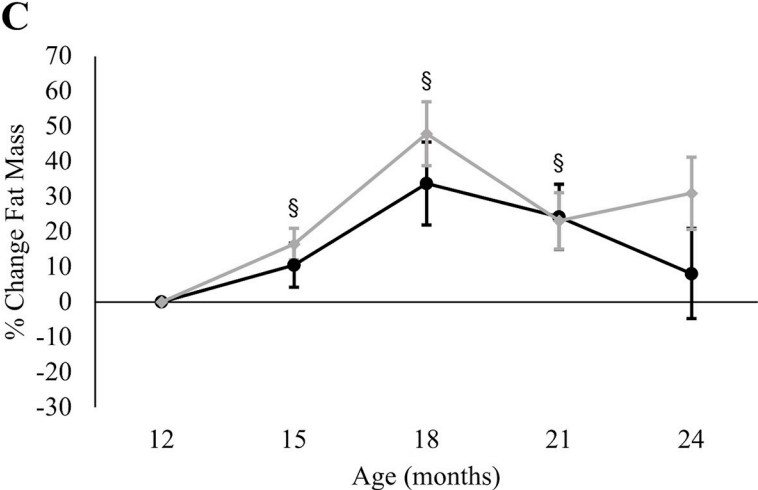

**Fig 2. Changes in total, lean, and fat mass in treated and control female mice.** (A) Compared to 12-month baseline, total body mass was significantly higher at all time points in the female group, and DOX treatment had no significant effect. (B) Lean mass was also significantly higher from 12-month baseline at all time points in the female, but more pronounced in DOX-treated females, a statistically significant difference compared to the control group. (C) Fat mass was significantly higher than 12-month baseline at 15, 18 and 21 months, and there was no effect of DOX treatment.

To show the main effect of time, DOX-treated and control values were pooled and the comparison to baseline indicated with § ($p < 0.05$); the asterisk (*) indicates significant difference between DOX-treated and control mice at specified time points ($p < 0.05$). Values are mean ± SEM.

Time and DOX treatment had a significant impact on fat mass, $F(2.66, 50.48) = 3.95$, $p = 0.016$ (Fig 3C). In the control group, fat mass increased until 18 months and then declined progressively until 24 months of age, but remained above baseline levels. In contrast, we found no significant change in fat mass in the DOX-treated group. The difference in fat mass between both groups was statistically significant (Bonferroni corrected) at 15 (mean difference from control 40.8%; 95% CI, 10.7% to 70.9%, $p = 0.011$), 18 (mean difference from control 43.6%; 95% CI, 16.5% to 70.7%, $p = 0.003$) and 24 months (mean difference between groups 43.6%; 95% CI, 5.4% to 81.8%, $p = 0.027$), but not significant at 21 months ($p = 0.06$).

Incidentally, toward the end of our study we began to leave heterozygous male littermates in cages with experimental mice (as companions) and some of these mice were included in body composition measurements for three months. The low numbers (DOX-control n = 7, Cre-control n = 2) preclude any statistical analysis, but the results are provided in Fig 4, and discussed in the section on limitations, below.

### Time course of changes in neuromuscular integrity

**MUNE.** There was no effect of DOX-treatment on MUNE recorded from the triceps surae muscle, $F(3, 105) = 0.542$, $p = 0.66$ (Fig 5A), but we found a significant two-way interaction of time and sex on MUNE, $F(3, 105) = 3.091$, $p = 0.038$. Follow-up comparisons (Bonferroni corrected) indicated that male mice experienced a greater reduction in MUNE than females at 15 months of age ($p = 0.04$) but there were no differences at any other time point. All mice exhibited a progressive decline in MUNE from baseline that became statistically significant by 21 months of age, and maintained at 24 months (both $p$-values < 0.001). Raw MUNE, SMUP, and CMAP values can be found in S1 Table.

**SMUP.** The average SMUP size represents an electrophysiological index of the average motor unit size (the average number of muscle fibers innervated by a single motor unit). SMUP was not modified by sex or DOX treatment, $F(2.84, 99.46) = 0.476$, $p = 0,69$ (Fig 5B). The progressive increase in SMUP mirrored the decline in MUNE in the triceps surae, as it was significantly higher by 21 months of age, and remained elevated at 24 months (both $p$-values < 0.001).

## Discussion

We set out to determine the effects of long-term post-developmental MSTN depletion on motor unit properties and body composition in aging male and female mice, with a longitudinal study between ages 12 and 24 months. We observed a significant decline in MUNE recorded from the triceps surae muscle between 15 and 21 months of age, and the SMUP size increased consistent with motor unit remodeling and reinnervation over the same time period. These age-related changes were not modified by a decrease in MSTN levels. There was, however, a combined effect of treatment and sex on changes in body composition. In females, reduction in MSTN levels resulted in a significant increase in lean mass compared to controls; in males, reduction in MSTN levels induced a decrease in fat accumulation compared to controls. These findings highlight a sexual dimorphic effect of DOX treatment/MSTN reduction on body composition in mice (most pronounced in the first three months of treatment), and provide evidence to suggest that post-developmental reduction in MSTN levels does not affect motor unit characteristics and does not prevent age-related neurodegenerative changes. This

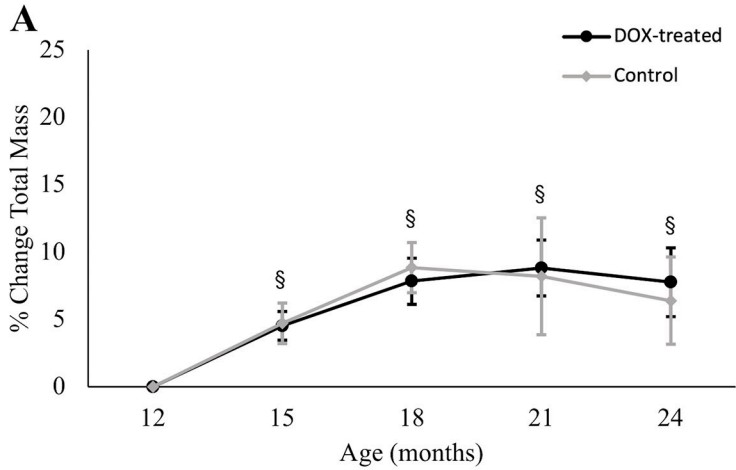

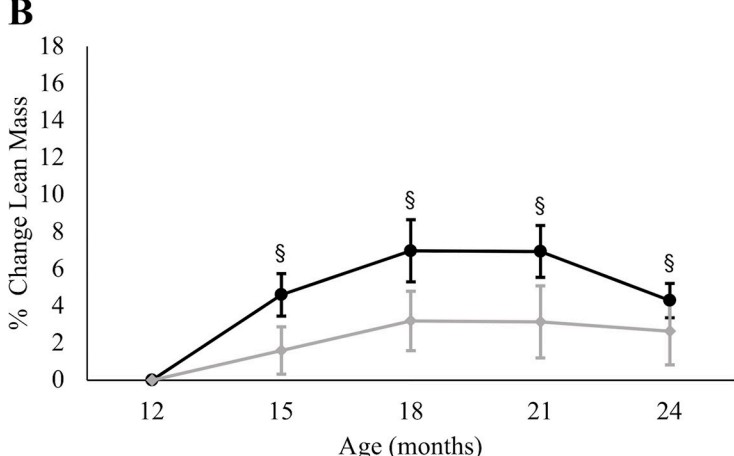

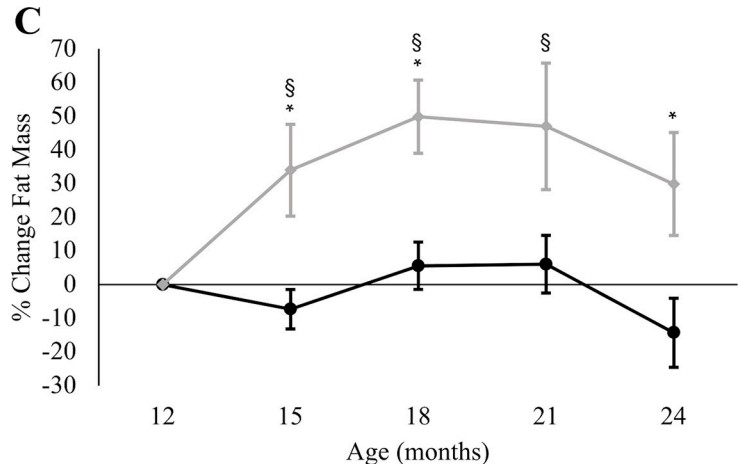

**Fig 3. Changes in total, lean, and fat mass in treated and control male mice.** (A) Compared to the 12-month baseline, total body mass in males was significantly higher at all time points, but there was no effect of DOX treatment. (B) Lean mass in males was also significantly higher at all time points, compared to 12-month baseline, with no effect due to DOX treatment. (C) The change in fat mass, however, was significantly higher than 12-month baseline at 15, 18 and 21 months of age. In addition, there was a treatment effect, and DOX-treated males had significantly less fat accumulation than male controls at 15, 18 and 24 months of age. To show the main effect of time, DOX-treated and control values were pooled and the comparison to baseline indicated with § ($p < 0.05$); the asterisk (*) indicates significant difference between DOX-treated and control mice at specified time points ($p < 0.05$). Values are mean ± SEM.

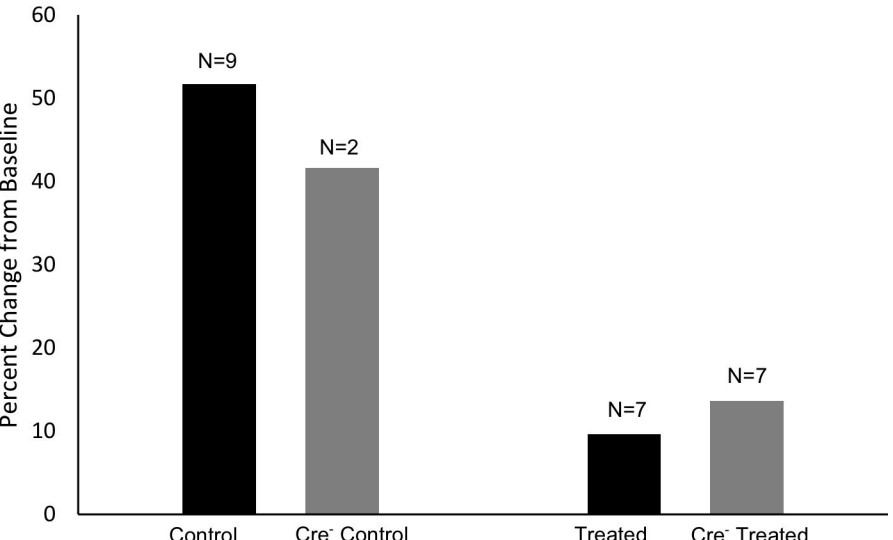

**Fig 4. Pilot data on the effect of 3 months administration of doxycycline.** Compared to baseline, male mice with (treated) and without (Cre⁻ treated) the Cre insertion that received DOX in the chow experienced a reduction in fat mass compared to baseline. By contrast, both mice positive (control) and negative (Cre⁻) for Cre that did not receive DOX exhibited a 40–50% increase in fat mass compared to baseline.

finding is critical in the evaluation of MSTN's therapeutic efficacy: given that age-related decline in muscle strength is more important for functional ability than muscle mass alone [44–46], muscle mass-enhancing therapies would likely be more successful if accompanied by increases in neural activation.

To our knowledge this is the first investigation into the effects of post-developmental MSTN reduction on electrophysiological motor unit characteristics in mice. Preclinical and clinical studies have shown similar patterns of motor unit number decline and remodeling during aging [41,47–49]. *Mstn*⁻/⁻ mice have a greater number of nerve fibers than wild type controls by three months of age [17,18], and age-related axonal loss is attenuated in the absence of MSTN [17]. By contrast, we found that reducing MSTN levels in adulthood had no impact on the age-dependent decrease in motor unit numbers. Overall, the lack of constitutive MSTN in mice results in phenotypic adaptations that are markedly different from those seen when MSTN levels are reduced in adulthood. For example, muscle mass has been reported to be 65–200% greater in *Mstn*⁻/⁻ mice than wild type controls [13,14,19,50], while muscle mass in post-development models is only 10–45% greater than controls [20–24]. Additionally, *Mstn*⁻/⁻ mice exhibit muscle fiber hyperplasia [18,19], greater distribution of type IIb/IIx fibers [14,51,52], lower mitochondrial content in the muscle [14,16], and accelerated muscle atrophy with disuse [53]. In contrast, post-developmental knock-out mice exhibit no hyperplasia [20,21,54], no change in fiber type distribution [23,26], no change in mitochondrial number [23], and attenuation of disuse-related atrophy [27].

The hyperplasia seen in the *Mstn*⁻/⁻ mouse is of particular importance in regard to neuro-muscular characteristics, as there must be some form of neural compensation in order to innervate the excess muscle fibers. Gay et al. [18] reported a 56% increase in the number of muscle fibers in the tibialis anterior of male and female *Mstn*⁻/⁻ mice, compared to controls, which was accompanied by a 23% increase in the number of motor units and a 27% increase in the number of muscle fibers innervated by a single motor unit. Taken together, the lack of hyperplasia in mice with post-developmental reduction in MSTN levels [20,21,54], and our

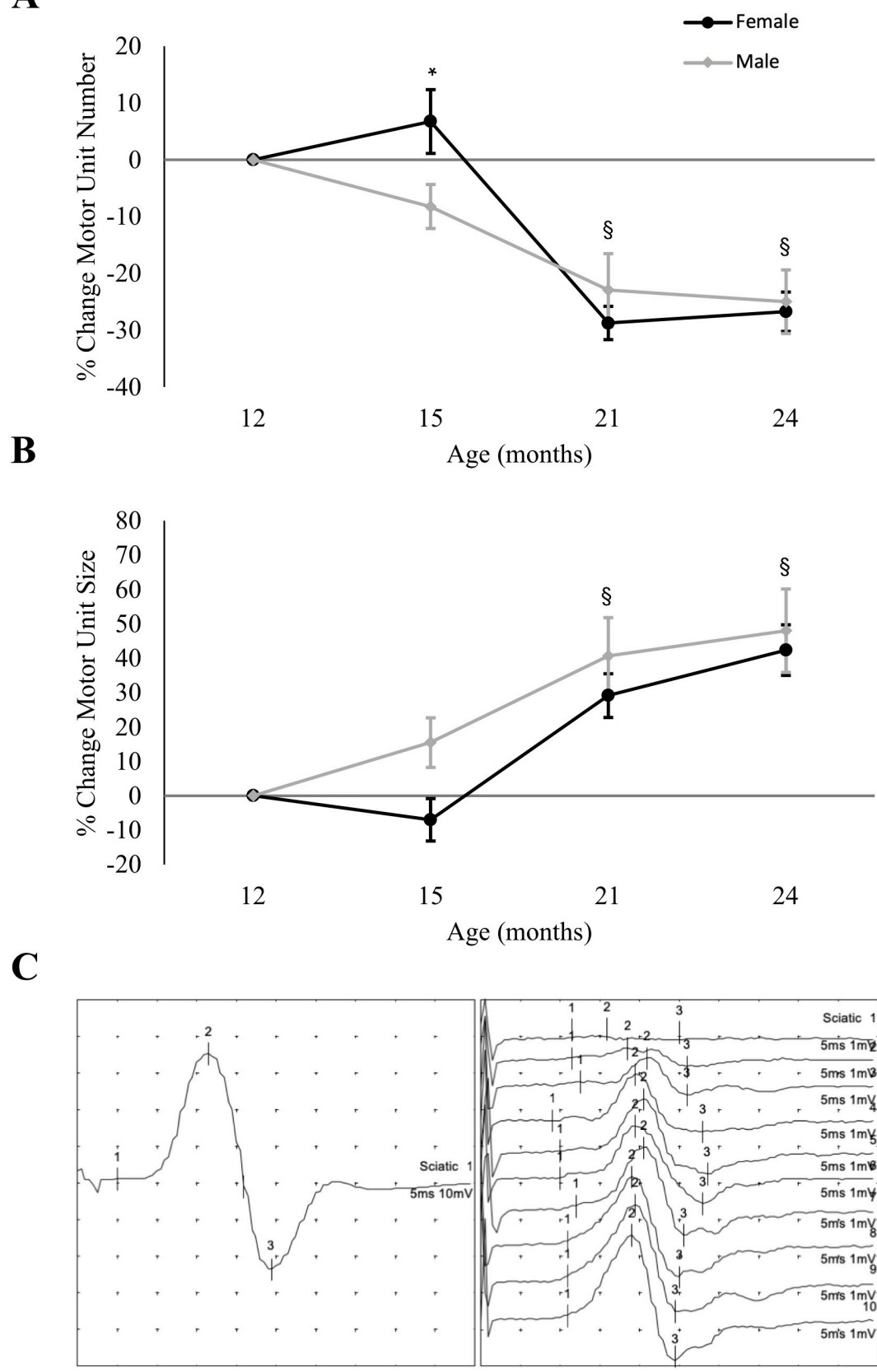

**Fig 5. Longitudinal changes in electrophysiological motor unit characteristics of male and female mice following long term DOX treatment.** DOX treatment had no significant effect on motor unit characteristics, and percent change in motor

unit number (A) and size (B) from the 12-month baseline was the result of time. Treatment groups were collapsed to better show this age-related impairment. (A) In both male and female mice there was a reduction in motor unit numbers by 21 months of age. Interestingly, at age 15 months female mice maintained their numbers while male mice had already experienced a reduction that was significantly different from females. From that time point, measures in both groups were significantly reduced from baseline. (B) As anticipated, increases in average SMUP size paralleled the decline in MUNE shown in A. At age 21 and 24 months the increase in SMUP size was significantly different from the 12-month baseline. (C) Representative trace of the electromyographic recordings (methodological details given in the text). M, male; F, female. § indicates significant difference from baseline measurements (effect of time) ($p < 0.05$); the asterisk (*) indicates significant difference between male and female mice at specified time point ($p < 0.05$). Values are mean ± SEM.

current results, suggest that the increase in the number of nerve fibers in the $Mstn^{-/-}$ mouse could be an indirect consequence of innervating a greater number of muscle fibers during development; MSTN would not exert a direct effect on motor unit characteristics.

MSTN reduction has been proposed as a potential therapeutic approach to counter several muscle wasting diseases, including sarcopenia [55], and recent investigations support this assertion. Camporez et al. [30] reported increased muscle mass and grip strength in response to four weeks of anti-myostatin antibody treatment initiated at 22 months of age. Similarly, Latres et al. [27] reported increased muscle mass and electrically-evoked force after only three weeks of myostatin monoclonal blocking antibody treatment initiated at 24 months of age. These two studies were performed exclusively in male mice. In wild type male and female mice, levels of the mature form of MSTN protein increase in muscle until three months of age, at which point they continue to increase in females but decline in males until eight months of age, the latest time point reported [56]. The decrease in mature MSTN in males is developmentally regulated by growth hormone (GH) [56], as the GH/IGF-1 axis is the major regulator of postnatal growth [57]. Male gonad related factors (i.e., testosterone) also affect postnatal development and skeletal muscle maintenance [58], and the combination of these factors result in sexually dimorphic muscle growth with age, in favor of males [56,58,59]. It follows, then, that males and females would adapt differently to experimental modulations of a variety of factors, including MSTN. However, most investigations into the effects of MSTN inhibition with aging have been single sex studies, performed only in males [27,28,30] or only in females [25,54].

Our results show that male and female mice respond differently to post-developmental MSTN reduction via DOX treatment, with females significantly increasing their lean mass, and males significantly reducing fat mass accumulation, compared to their respective controls. The reason for the females' greater increase in lean mass is unclear, but may be related to MSTN reduction counteracting barriers to muscle growth in females (e.g., lower GH/IGF-1 levels and elevated MSTN levels) [56,58,59], while males may already be near optimal muscle growth levels prior to the reduction. Another possibility is that the differences in circulating levels of certain estrogens in females can produce anabolic effects in the absence of myostatin [60]. In humans, basal rate of muscle protein synthesis is increased in older females, compared to older men and young men and women, which may also be linked to concentration of estrogens [61]. Ovariectomized rats exhibit increases in lean mass compared to controls, but this effect disappears when ovariectomized rats are supplemented with estradiol or progesterone, suggesting that these estrogens prevent muscle growth [62]. Post-menopausal females may then have a greater potential to increase muscle mass with certain interventions, compared to younger adults or older males who do not see similar changes in female gonad related factors.

Reisz-Porszaz et al. [43] also reported a dimorphism in body composition of young male and female transgenic mice overexpressing MSTN. In that study, male transgenic mice had significantly lower muscle mass than their controls, but there were no differences in muscle

mass between female transgenic and female controls. Additionally, male transgenic mice had greater fat mass than male controls, but fat mass in female transgenic mice was not significantly different from female controls (food consumption was not reported) [43]. Based on those results, it can be suggested that males are less responsive to the anabolic effects of reduced MSTN levels than females, but are more responsive to the catabolic effects of elevated MSTN levels. Nonetheless, there may be other unknown mechanisms in female mice that override the effect of MSTN on fat mass.

We also considered the possibility that DOX treatment could impact food consumption, and thus affect body composition. However, DOX-treated females consumed less food, relative to body weight, than their controls while increasing lean mass to a greater extent. Additionally, DOX-treated males consumed more food than their controls, but had significantly less fat mass accumulation. The response to decreased food consumption in treated females and increased food consumption in treated males is counter-intuitive, and again highlights the sexual dimorphic effect of MSTN reduction and DOX treatment in mice.

The long-term effects of MSTN deficiency have been studied in $Mstn^{-/-}$ mice up to two years of age [50,63,64], but post-developmental MSTN reduction has been reported only in short-term studies, ranging from 3–4 weeks [27,30,54] to 3–4 months [20,25,26]. A study by Personius et al. [23] recorded extensor digitorum longus contractile properties 9 months after post-developmental, tamoxifen-induced *Mstn* gene deletion (initiated at 4 months of age), but did not report values beyond the 13 month-old time point. Our study initiated the reduction in MSTN levels in 12-month-old mice and tracked changes in body composition over the following 12 months. Compared to controls, female DOX-treated mice had significant increases in lean mass at all time points, with the greatest rate of divergence between 12 and 15 months of age (5.3% difference). Lean mass values continued to diverge at each subsequent time point, but to a lesser extent, increasing by an additional 3.5% between 15 and 24 months. In males, DOX treatment had a powerful and dramatic effect, with 40.8% less fat accumulation than controls within the first 3 months of treatment (between 12 and 15 months of age), and a muted effect (an additional 2.8% difference) in the following three months (between 15 and 18 months of age). Taken together, these findings indicate that the most robust body composition adaptations to post-developmental MSTN reduction via DOX treatment occur within the first three months of treatment in aging mice, with minor additional changes following prolonged (> 6 months) treatment.

## Limitations

In the present study MUNE was used to allow repeat assessment of the estimated number of motor axons functionally innervating the triceps surae. There is inherent variability in MUNE as the size of only a limited number of single motor units can be assessed (i.e. SMUP). While recording more increments during the incremental MUNE technique could theoretically improve the precision and accuracy of the estimated average SMUP size, this benefit is offset by the increased likelihood of alternation which can falsely decrease SMUP size and inflate MUNE values [65]. Therefore, incremental MUNE is performed by averaging only 10 increments [66]. While we did not perform test-retest experiments in our study, in our experience test-retest coefficient of variability for MUNE studies in rodents is in the range 5–10% (unpublished observations). Another limitation of MUNE is that there are no other gold standards for validating the number of functional motor units that are innervating a particular muscle. A characteristic of CMAP is that its amplitude does not reflect the size of the muscle, rather it is a measure of motor unit density, which is why MUNE is an estimate [66].Anatomical counts can offer some insight, but these are limited to labeling techniques that are also not without

caveats [67]. Very few studies have investigated both electrophysiological MUNE and anatomic counts of motor neurons. In these studies, histological counts are discordant with electrophysiological MUNE, but relative values are consistent, and MUNE correlates with other clinical measures of motor unit decline [66]. Nevertheless, future studies could include anatomical counts of spinal motor neurons or peripheral nerve axons, acknowledging that those techniques would not allow repeated measures.

We chose the current mouse model as the most appropriate to study interactions between muscle and motoneurons in mice, with the understanding that Cre activation in the absence of DOX would be negligible [35]. The experimental animal was the result of breeding two separate transgenic models with different backgrounds (see methodology), and as such there was no appropriate wild type mouse to serve as a control. Although heterozygous littermates have been recommended as controls [68], we were aware that another group using a similar, tamoxifen-inducible model had reported a reduction in MSTN mRNA expression by four months of age in *untreated* homozygous mice [20]. Thus, there was a possibility that heterozygous controls and homozygous experimental mice would have vast differences in MSTN levels at the beginning of testing (12 months of age). In light of this, we chose to use the mice with the homozygous floxed genotype, (i.e., testing positive for the *tTA*, *CRE*, and *MSTN-3 FL* probes) for both the experimental and control groups, thus leaving DOX treatment (and subsequent Cre activation to fully inactivate *Mstn*) as the only difference between the two groups.

The effects of chronic DOX treatment on body composition in mice had not been published at the start of the study (2013); however, in 2017 Wang et al. [69] reported decreased fat mass accumulation with ten weeks of DOX treatment in male *db/db* mice. A major benefit of MSTN reduction therapy is improved body composition, as increased lean mass and decreased fat mass has been reported in constitutive knock-out mice [70,71], while post-developmental models indicate that MSTN reduction prevents fat mass accumulation but does not reduce existing fat mass levels [24,72,73]. DOX treatment, therefore, is potentially a major confounding factor when body composition changes are primary endpoints of a study. We report unique adaptations to DOX treatment/MSTN reduction between males and females, with DOX-treated females increasing lean mass to a greater extent than controls (with no differences in fat mass), and DOX-treated males demonstrating reduced fat mass accumulation compared to controls (with no difference in lean mass). However, it is impossible to know if these sex-specific differences are due to DOX, MSTN reduction, or a combination of the two. Thus, we report, albeit preliminary, body composition measurements obtained from heterozygous littermates that were kept as companions in cages with experimental mice (Fig 4). When this body composition data is brought into consideration, we conclude that the reduced accumulation of fat mass in males is likely due to DOX administration, and not MSTN reduction.

Our use of a transgenic mouse model with a DOX-inducible, Cre-mediated recombination system specific to exon 3 of the *Mstn* gene presented an advantage by avoiding potential compensatory mechanisms that occur during development in the constitutive deletion model. However, it also carries some potential limitations, as the inducible Cre can be "leaky". In fact, a previous study using a tamoxifen-inducible model reported a 50–90% reduction in myostatin mRNA before inducing Cre activation (compared to controls with no Cre transgene) by four months of age [20]. We found a similar effect in our model. Blood samples from four male mice lacking the Cre gene were also added at the end of the study, and their MSTN plasma levels at 24 months of age were >2-fold greater than the non-treated control mice. We conclude that there is substantial Cre activation in the absence of DOX treatment. This may have affected our results by increasing variability among subjects and reducing the size of the treatment effect, since both groups demonstrated substantial reductions in MSTN. In other words, we found a 37% reduction compared to a control group that also had reduced MSTN levels

due to the 'leaky' construct. Compared to the Cre⁻ group, the reduction in MSTN levels would have been closer to 80%. However, this finding should be interpreted with caution because the reference group was not adequately powered (n = 4) [59]. Nonetheless, despite this leakiness, the 37% reduction in MSTN levels were accompanied by significant differences in body composition over time *relative to each group's baseline* in response to DOX treatment. This pronounced effect strongly suggests that, in adult mice, MSTN is involved in the regulation of body composition, in agreement with Lozier et. al [74], and perhaps in energy homeostasis. Lastly, while our experimental model targeted the deletion of skeletal muscle-specific *Mstn*, it should be noted that MSTN can be produced by other organs, in different levels, and in particular the tongue has been found to contain high levels of MSTN mRNA [75].

## Conclusion

Long-term (12 months) post-developmental decrease in MSTN levels via DOX treatment starting at age 12 months did not affect motor unit characteristics or attenuate the motor unit degeneration that accompanies aging. However, the manipulation did result in body composition adaptations that differed between males and females. DOX-treated female mice significantly increased their lean mass compared to controls, and DOX-treated male mice had significantly less body fat accumulation than their controls. MSTN appears to have differing effects on male and female mice, which may translate to human applications. Our results strongly suggest that future studies investigating the role of MSTN should include both male and female subjects to allow for a comparison. Furthermore, we show that the most robust effects of MSTN reduction occur in the first three months after treatment initiation. Future studies investigating the effects of MSTN may consider limiting treatment duration within this range.

Our results strongly suggest that when the experimental design requires using this or similar Tet-ON model with DOX as the inducer, Cre leakiness and the independent effect of DOX treatment must be evaluated with additional controls. One possible approach to parse out the unique effects of DOX treatment and MSTN reduction is to use both heterozygous and homozygous animals, with both groups split into single sex, DOX-treated and untreated cages. Ensuring enough statistical power with sufficiently large N per group (8 groups in the scenario here suggested) would also be required, which would increase considerably the number of animals used for the study. An alternative to reduce the number of animals would be to include a model verification at each stage of the study, thus obtaining longitudinal data on the effects of DOX versus levels of MSTN. However, obtaining multiple muscle biopsies or blood collections from the same animal are frequently discouraged by IACUCs due to the repeated distress inflicted in the mice. This was the reason that led us to collect blood serum for model verification only at the end of our study. In any case, the advent and widespread use of CRISPR/Cas9 technology calls into question the use of Cre-mediated recombination in mice [69].

MSTN reduction remains a potential therapeutic option to treat sarcopenia and dynapenia, though it is no longer the only biologically-targeted option. GDF11, another member of the Transforming Growth Factor-$\beta$ superfamily, is also a negative regulator of muscle mass that has therapeutic potential [76]. Additionally, follistatin is a binding protein capable of blocking MSTN activity, ultimately increasing muscle mass [77]. Follistatin also appears to block other ligands with activity similar to myostatin, and increasing follistatin levels in *Mstn*⁻ᐟ⁻ mice doubles the effect of *Mstn* gene deletion, resulting in a 4–6 fold increase in muscle mass over wild type controls [77]. While encouraging, a better understanding of the efficacy and safety of these therapeutic strategies, as well as teasing out primary targets versus secondary effects, will be necessary if they are to be used to counter disease- and age-related muscle loss. However,

the lack of neural adaptations in response to reduced MSTN suggests that these strategies may not produce the desired functional improvements that should accompany the increased muscle mass unless the neural system is also targeted.

## Supporting information

**S1 Table. Food consumption, body mass, and motor unit characteristics between 12 (baseline) and 24 months of age.** Values are mean ± SEM of 10 male and 12 female control, and 11 male and 14 female DOX-treated mice. MUNE (motor unit number estimation) values are an estimate of the number of motor units in the right triceps surae muscle. SMUP (single motor unit potential size) values are the average size of motor units in the right triceps surae muscle, quantified as peak-to-peak distance in μV. μV, microvolts; g, grams.
(DOCX)

## Acknowledgments

The authors would like to thank Neena McIlwaine, Haley Appelmann, Nathanial Bulle, Caitlyn Loveridge, and Dakota Brockway for their contributions in animal care and data collection.

## Author Contributions

**Conceptualization:** Sonsoles de Lacalle.

**Data curation:** Sophia C. Mort, Sonsoles de Lacalle.

**Formal analysis:** Dallin Tavoian, Sonsoles de Lacalle.

**Funding acquisition:** W. David Arnold, Sonsoles de Lacalle.

**Investigation:** Dallin Tavoian, W. David Arnold, Sophia C. Mort, Sonsoles de Lacalle.

**Methodology:** W. David Arnold, Sonsoles de Lacalle.

**Project administration:** Sonsoles de Lacalle.

**Resources:** Dallin Tavoian, Sonsoles de Lacalle.

**Supervision:** Sonsoles de Lacalle.

**Validation:** Dallin Tavoian.

**Writing – original draft:** Dallin Tavoian, Sophia C. Mort, Sonsoles de Lacalle.

**Writing – review & editing:** Dallin Tavoian, W. David Arnold, Sonsoles de Lacalle.

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
