## [Decision Letter · Decision Letter 0]

8 Aug 2019

PONE-D-19-16707

Sex differences in body composition but not neuromuscular function following long-term, doxycycline-induced reduction in circulating levels of myostatin in mice

PLOS ONE

Dear Dr de Lacalle,

Thank you for submitting your manuscript to PLOS ONE. After careful consideration, we feel that it has merit but does not fully meet PLOS ONE’s publication criteria as it currently stands. Therefore, we invite you to submit a revised version of the manuscript that addresses the points raised during the review process.

For the non-expert reader please provide a more detailed description of how motor unit number was derived from EMG data. Similarly explain how average motor unit size was estimated. The results show % changes but not your estimate of the absolute number of motor units in the muscle. It would be helpful to provide your estimates of the number of motor unis in the muscle to aid comparison with current published estimates (if any) and/or as a baseline estimate for future studies. Also, some discussion of the limitations of the MUNE method in mice (e.g. what is known about error arising from repeated MUNE estimates for the same muscle in the same animal) and the degree to which this might limit your ability to discern potential treatment effects.

Serum myostatin levels were reduced by 37% in the DOX treated animals. Given the muscle knockdown which tissues are producing the remaining 63% of blood myostatin (or is the muscle knockdown incomplete?), how does the blood myostatin level compare to that of preceding studies. Do we know that the 37% reduction is biologically meaningful and if so what is the reason to believe this? Add discussion to clarify.

Consider adding references to published data on motoneuron loss in old age and discuss briefly how these might (or might not) correlate with your data on reduction in motor unit number. 

We would appreciate receiving your revised manuscript by Sep 22 2019 11:59PM. To enhance the reproducibility of your results, we recommend that if applicable you deposit your laboratory protocols in protocols.io, where a protocol can be assigned its own identifier (DOI) such that it can be cited independently in the future. For instructions see: http://journals.plos.org/plosone/s/submission-guidelines#loc-laboratory-protocols

We look forward to receiving your revised manuscript.

Kind regards,

William D Phillips

Academic Editor

PLOS ONE

Journal Requirements:

Reviewers' comments:

Reviewer's Responses to Questions

**Comments to the Author**

1. Is the manuscript technically sound, and do the data support the conclusions?

Reviewer #1: Partly

2. Has the statistical analysis been performed appropriately and rigorously? 

Reviewer #1: Yes

3. Have the authors made all data underlying the findings in their manuscript fully available?

Reviewer #1: Yes

4. Is the manuscript presented in an intelligible fashion and written in standard English?

Reviewer #1: Yes

5. Review Comments to the Author

Reviewer #1: The manuscript investigates the impact of long-term reduction of myostatin on number and size of motor units in mouse a hindlimb muscle group, along with measures of body composition (esp. fat and lean mass). The rationale is justified since countermeasures against age-related loss of muscle mass are sought, and whilst myostatin reduction has some of the features of a viable treatment, it has not been investigated in a model in which treatment is long term and onset is middle age. There are several findings. The first is that myostatin reduction has no impact on the well described reduction in motor unit number or increase in motor unit size that typically accompany advancing age. The second is that myostatin reduction results in alteration of body composition during the early treatment phase, but with effects that manifest differently in male and female mice. The third is that the DOX administration required to trigger the myostatin reduction may itself have a direct impact on the measured body composition parameters.

The methods are generally described in adequate detail, though I had some difficulty with the electromyographic methodology. I presume that motor unit number estimation derives from progressive increments in the size of the CAP on increasing stimulus voltage. Fig 5C gives an example trace, but it is difficult to reconcile the method, the sample traces, and the data in the supporting table which states that at baseline the muscles had of the order of 350 motor units. I would appreciate a slightly more detailed description of the method with clear indication of how motor unit number was derived from EMG data.

Serum myostatin levels were reduced by 37% in the DOX treated animals. Whilst this had a likely impact on some of the subsequently measured parameters, we are given no indication what other (non-muscle) sources of myostatin might be unaffected by the treatment (where did the myostatin come from?), how this myostatin level compares to that of preceding studies, or whether/how this reduction is biologically meaningful.

The discussion of outcomes relating to motor unit number and size are interesting and insightful. I agree that any therapeutic intervention needs to ensure preservation or restoration of motor drive, so evidence that myostatin reduction has no impact on loss of neural input is both new and valuable information. I would like to see slightly more in-depth treatment of the matter of neuron loss, the authors could add reference to published data on motoneuron loss in old age to correlate with their data on reduction in motor unit number. To my mind this would strengthen validity of the experimental approach, especially given my comments above regarding the relatively obscure derivation of motor unit number from EMG data.

The changes in body composition are described, analysed, and documented in adequate detail, though the interpretation of this dataset is problematic. The authors are honest and open in their acknowledgement that the results are surprising and cannot be fully explained with currently available data. It is difficult to know what to make of the body composition results, and the discussion of these findings is consequentially speculative. The authors concede that further data is needed before a valid conclusion can be drawn, and they also raise the prospect that the DOX treatment paradigm may be having an unpredictable direct impact on the outcomes. The leads to a relatively lengthy description of limitations and a strong indication that the authors consider the methodology to be questionable.

6. PLOS authors have the option to publish the peer review history of their article (what does this mean?). If published, this will include your full peer review and any attached files.

Reviewer #1: No

---

## [Decision Letter · Decision Letter 1]

26 Sep 2019

PONE-D-19-16707R1

Sex differences in body composition but not neuromuscular function following long-term, doxycycline-induced reduction in circulating levels of myostatin in mice

PLOS ONE

Dear Dr de Lacalle,

Thank you for submitting your manuscript to PLOS ONE. After careful consideration, we feel that it has merit but does not fully meet PLOS ONE’s publication criteria as it currently stands. Therefore, we invite you to submit a revised version of the manuscript that addresses the points raised during the review process.

Please address the reviewer's comments (detailed below) specifically:

1. Check and correct as necessary the CMAP values in Table S1 (which would normally be measured in microvolts).

2. Provide a fuller discussion of the differences between your motor neuron number estimates by MUNE and other published estimates of the motor neuron pool size (by other methods). It is important for readers to be aware of differences in motor neuron numbers that may arise through different methodologies and of potential sources of error in such estimates.

We would appreciate receiving your revised manuscript by Nov 10 2019 11:59PM. To enhance the reproducibility of your results, we recommend that if applicable you deposit your laboratory protocols in protocols.io, where a protocol can be assigned its own identifier (DOI) such that it can be cited independently in the future. For instructions see: http://journals.plos.org/plosone/s/submission-guidelines#loc-laboratory-protocols

We look forward to receiving your revised manuscript.

Kind regards,

William D Phillips

Academic Editor

PLOS ONE

Reviewers' comments:

Reviewer's Responses to Questions

**Comments to the Author**

1. If the authors have adequately addressed your comments raised in a previous round of review and you feel that this manuscript is now acceptable for publication, you may indicate that here to bypass the “Comments to the Author” section, enter your conflict of interest statement in the “Confidential to Editor” section, and submit your "Accept" recommendation.

Reviewer #1: (No Response)

2. Is the manuscript technically sound, and do the data support the conclusions?

Reviewer #1: Yes

3. Has the statistical analysis been performed appropriately and rigorously? 

Reviewer #1: Yes

4. Have the authors made all data underlying the findings in their manuscript fully available?

Reviewer #1: Yes

5. Is the manuscript presented in an intelligible fashion and written in standard English?

Reviewer #1: Yes

6. Review Comments to the Author

Reviewer #1: In my original review I indicated that the question was interesting, and that the approach had the potential to add new insight into possible value of myostatin therapy as a countermeasure for age-related loss of muscle mass. My feelings about those issues remain unchanged. I made a number of comments and suggestions for clarification in a revised manuscript, and the authors have made minor changes to address each of those in turn. In my view, two issues remain. First, data has been added to Table S1, but there is an error in data presentation that needs correction. CMAP amplitude is stated as about 80mV, with SMUP at 232mV. It is not possible for these data to be correct and to lead to the stated MUNE. Almost certainly, the SMUP data should be in microvolts, or listed as ~0.2mv instead of ~200mV.

Second, I had asked for a discussion of the way in which the current data compares with previously published data on motor unit number changes with age. The authors have added a brief statement (with references) indicating that other studies have shown reduction in motor unit number with age, but have not added any discussion and appear to have missed the main point of my comment. It may be that my comment was obscure and for that I apologise and I will now try to be more direct. The authors acknowledge that the technique provides only an estimate of motor unit number, so our confidence in the accuracy of the estimate would be enhanced if the numbers were similar to estimates using other techniques, ie how does this result correlate with other methods that have assayed motor neuron (or motor unit) number to the triceps surae group in the mouse? For example, in their classical work in 1981, McHanwell & Biscoe performed motor neuron counts using histological and retrograde tracing techniques. Their work indicated an average triceps surae motor pool size of 154 neurons (range 116-193). This and other work (cited in the 1981 paper) indicated a total lumbar motoneuron count in the range of 1700-2200 motoneurons (and more recent studies are consistent with a total lumbar motoneuron number around 2000-2200 in young mice), so the triceps surae pool likely represents a bit less than 10% of the total lumbar motor neuron count in the mouse. In the current study, results indicate that about 350 motoneurons innervate triceps surae muscles, suggesting (using McHanwell's proportions) a total lumbar motoneuron number approaching 4000 which, to my knowledge, no study has ever reported. So, my previous comment was asking the authors to reconcile this discrepancy between their TS motor unit number estimates (~350) and those of other authors (~150), acquired using other techniques. Note that I am not questioning or disputing the outcome showing a decrease in motor unit number with age, and I am not suggesting that any particular dataset is correct or incorrect, but (in my view) the discrepancy between the datasets is big enough to require comment.

7. PLOS authors have the option to publish the peer review history of their article (what does this mean?). If published, this will include your full peer review and any attached files.

Reviewer #1: No

---

## [Author Response · Author response to Decision Letter 1]

18 Oct 2019

We respond to the reviewer's concerns in the attached file.

---

## [Editor Report · Decision Letter 2]

1 Nov 2019

Sex differences in body composition but not neuromuscular function following long-term, doxycycline-induced reduction in circulating levels of myostatin in mice

PONE-D-19-16707R2

Dear Dr. de Lacalle,

We are pleased to inform you that your manuscript has been judged scientifically suitable for publication and will be formally accepted for publication once it complies with all outstanding technical requirements.

With kind regards,

William D Phillips

Academic Editor

PLOS ONE
---

## [Editor Report · Acceptance letter]

12 Nov 2019

PONE-D-19-16707R2 

Sex differences in body composition but not neuromuscular function following long-term, doxycycline-induced reduction in circulating levels of myostatin in mice 

Dear Dr. de Lacalle:

I am pleased to inform you that your manuscript has been deemed suitable for publication in PLOS ONE. Congratulations! Your manuscript is now with our production department. 

With kind regards,

on behalf of

Dr. William D Phillips 

Academic Editor

PLOS ONE